# High-Sensitivity PtSe_2_ Surface Plasmon Resonance Biosensor Based on Metal-Si-Metal Waveguide Structure

**DOI:** 10.3390/bios12010027

**Published:** 2022-01-06

**Authors:** Zhitao Lin, Yiqing Shu, Weicheng Chen, Yang Zhao, Jianqing Li

**Affiliations:** 1Faculty of Information Technology, Macau University of Science and Technology, Avenida Wai Long, Taipa, Macao 999078, China; 1809853gii30001@student.must.edu.mo (Z.L.); 1909853yii30001@student.must.edu.mo (Y.S.); 2Guangdong-HongKong-Macao Joint Laboratory for Intelligent Micro-Nano Optoelectronic Technology, Foshan University, Foshan 528225, China; chenwch@fosu.edu.cn; 3Weihai City Key Laboratory of Photoacoustic Testing and Sensing, Harbin Institute of Technology (Weihai), Weihai 264209, China; zhao.yang@hit.edu.cn; 4Zhuhai MUST Science & Technology Research Institute, Zhuhai 519000, China

**Keywords:** surface plasmon resonance, biosensor, PtSe_2_, waveguide

## Abstract

PtSe_2_ as a novel TMDCs material is used to modify the traditional SPR biosensors to improve the performance. On this basis, this research proposes a metal-Si-metal waveguide structure to further improve the performance of the biosensor. In this study, we not only studied the effects of waveguide structures containing different metals on the performance of biosensor, but also discussed the performance change of the biosensor with the change of PtSe_2_ thickness. After the final optimization, a BK7-Au-Si-Au-PtSe_2_ (2 nm) biosensor structure achieved the highest sensitivity of 193.8°/RIU. This work provides a new development idea for the study of SPR biosensors with waveguide structures in the future.

## 1. Introduction

Surface plasmon resonance (SPR) found at the interface of two media (such as dielectric and metal) with opposite signs of dielectric constant can be excited by Otto or Kretschmann prism coupling model [1,2]. In recent years, SPR biosensors have been applied in food safety [3,4], environmental detection [5], medical diagnosis [6,7,8] and other fields, due to their outstanding performance of high detection accuracy, real time [9,10] and label free [11]. Traditional SPR biosensor structures usually contain glass prisms and noble metals which are in direct contact with biomolecules. However, it is found that the metals other than gold are easy to be oxidized by air, which may reduce the performance of the biosensor. At the same time, the adsorption capacity of metals for biomolecules is limited, which can greatly limit the sensitivity of biosensors. One of the most effective ways to lift the limitations of traditional SPR biosensors is to coat the biosensor structures with two-dimensional (2D) materials [12,13]. Most 2D materials have excellent time stability [14], which protects the metals in biosensors from oxidation, so as to improve the overall durability. In addition, 2D materials have large specific surface area [15], which can boost the adsorption capacity of biosensors.

Transition metal dichalcogenides (TMDs) are incrementally applied in biosensors for their excellent optical, electrical and mechanical properties [16]. Moreover, it can form heterostructures with other 2D materials to further enhance the detection performance of biosensors [17,18]. Among them, platinum diselenide (PtSe_2_) which is group-10 2D TMDs has adjustable band gap from 0 eV (bulk) to 1.2 eV (monolayer) [19,20]. Monolayer PtSe_2_ shows the characteristics of semiconductor, while bulk PtSe_2_ shows the characteristics of type-II Dirac semi-metal [21]. Additionally, bulk PtSe_2_ shows excellent physical properties, such as magnetoresistance, anomalous Hall effect, etc. [22]. Moreover, theoretical calculation predicted that PtSe_2_ has anisotropic carrier mobility (x direction 3250 cm^2^ V^−1^ S^−1^ and y direction 16,300 cm^2^ V^−1^ S^−1^) at room temperature [23]. The synthesis of PtSe_2_ requires a layer of Pt to be deposited on the substrate in advance, so that PtSe_2_ can be synthesized in large-scale [24]. Furthermore, some studies have shown that PtSe_2_ has high chemical stability and less toxicity suitable for applications in continuously working biosensors [25]. Therefore, PtSe_2_-based biosensors have attracted research interest for performance improvement [26].

Different metals exhibited different performance in biosensors. Au-based biosensors have higher sensitivity, while Ag-based biosensors have narrower full width at half maximum of reflection curve [26]. In this work, PtSe_2_ as a molecular recognition element has been used to optimize the traditional SPR biosensor. On the one hand, it can improve the antioxidant capacity of biosensors with the addition of high-stability materials. On the other hand, it can improve the adsorption capacity of biomolecules, so as to improve the performance of biosensor. In addition, a silicon layer with a certain thickness can be inserted into the middle of the metal layer to form a metal-Si-metal waveguide structure. In order to increase the propagation distance of surface plasma wave, it is an effective method to add planar waveguide to eliminate the metal damping [27]. Therefore, many planar waveguide structures have been used to improve the performance of biosensors [28,29,30]. The thickness of Si layer can be optimized in the analysis of data. After determining the structural parameters of the proposed biosensor, the effects of different refractive indexes of the sensing medium and different wavelengths of incident light on the performance of the biosensor have been discussed. Finally, the thickness of PtSe_2_ has been optimized to achieve the optimal biosensor performance. At the same time, the effects of different waveguide structures which composed of different metals on the performance of PtSe_2_-based biosensors are compared.

## 2. Calculation Models and Methods

The structure of the proposed PtSe_2_-based biosensor is shown in Figure 1. BK7 glass with low refractive index works as a coupling prism. Next, there is a metal-Si-metal waveguide structure with a silicon layer inserted in the middle of the metal layers. Additionally, the biomolecular recognition element (PtSe_2_ layer) was coated on the waveguide structure surface. Meanwhile, the thickness of Au, Si (d_3_) and PtSe_2_ (d_5_) layers are optimized in the subsequent data discussion. The refractive index of BK7 glass can be expressed by the following formula [31]:$$n_{BK7}={\left(\frac{1.03961212\lambda^{2}}{\lambda^{2}-0.00600069867}+\frac{0.231792344\lambda^{2}}{\lambda^{2}-0.0200179144}+\frac{1.01046945\lambda^{2}}{\lambda^{2}-103.560653}+1\right)}^{1/2},$$
where the wavelength *λ* of incident light is in the unit of μm. Drude-Lorentz model is the most effective model to express the refractive index of metals [32]:$$n_{metal}=\sqrt{1-\frac{\lambda^{2}\lambda_{c}}{\lambda_{p}^{2}\left(\lambda_{c}+i\lambda \right)}},$$
where *λ_c_* is the collision wavelength and *λ_p_* is the and plasma wavelength, respectively. The collision and plasma wavelengths of involved metals are shown in Table 1.

The refractive index of the Si layer also changes with different incident wavelengths [33]: $n_{Si}=A+A_{1}e^{-\lambda /t_{1}}+A_{2}e^{-\lambda /t_{2}},$ where A = 3.44904, A_1_ = 2271.88813, A_2_ = 3.39538, t_1_ = 0.058304 and t_2_ = 0.30384. The unit of incident light wavelength (λ) is μm. According to previous reports, the extracted dielectric constant real part (ε_1_) and imaginary part (ε_2_) of PtSe_2_ are shown in Figure 2 [34].The working wavelength of the biosensor is chosen as 633 nm. Additionally, the refractive index of the sensing medium (n_s_) on the biosensor is chosen as 1.33.

The transmission matrix method is used to analyze the N-layer structure incident by TM-polarized light [35]. Therefore, the reflectance is defined as $R_{p}={\left|r_{p}\right|}^{2},$ and *r_p_* represent the reflection coefficient [36]. The sensitivity, also an important parameter, is expressed as $S_{n}=\Delta \theta_{SPR}/\Delta n_{s},$ where ∆θ*_SPR_* represents the shift of resonance angle with the small change of refractive index of the sensing medium (∆*n_s_*) [37]. The figure of merit (FOM) is the ratio of sensitivity to the full width at half maximum (FWHM), which can be expressed as FOM = S/FWHM [26].

## 3. Results and Discussions

The traditional Kretschmann biosensor shown in Figure 3a uses a single metal layer to excite SPR. The minimum reflectance and sensitivities have been discussed with different Au layer thicknesses. Additionally, the value of the minimum reflectance represents the utilization of the incident light energy. Although the sensitivity of the traditional biosensor enhances with the increase in the Au layer thickness, the thickness of the Au layer is selected as 50 nm after trading off. Additionally, the sensitivity value of traditional Au-based SPR biosensor at this thickness is 136.8°/RIU. However, this sensitivity value is still insufficient for high-performance biosensors. On the bases of traditional SPR biosensor structure, a 2 nm PtSe_2_ layer as a biomolecular recognition element is inlaid between the Au layer and the sensing medium (Figure 3b), and the sensitivity of this biosensor is improved to 156.6°/RIU. Figure 3c shows the reflectance curves varying with the incident angles of biosensor with Au-Si-Au waveguide structure for further improvement sensitivity. The metal thickness on both sides of the waveguide is set to 25 nm. Additionally, in the later discussion, the thickness of the metal in the waveguide maintains this value. The resonance angle position moves towards the large angle direction with the increase in Si layer thickness, and the FWHM of reflectance curves gradually broadens. After optimization, it is found that the reflectance at the resonance angle increases sharply when the Si layer is thicker than 13 nm (Figure 3d). The sensitivity of the proposed biosensor model reaches a maximum of 193.8°/RIU at this configuration. This shows that the proposed structure has a prominent effect on improving the performance of biosensor. Therefore, the thickness of the Si layer can be fixed at 13 nm in the subsequent discussion.

The detection range is an important parameter to measure the performance of a biosensor. In Figure 4, the performance of the innovative biosensors is measured in different refractive indices of the sensing medium. The curves of reflectance versus incident angle under different refractive indices of the sensing medium are presented in Figure 4a. In this figure, the Au-Si-Au waveguide structure has been used in the biosensor. With the increases in the sensing medium refractive index (n_s_ = 1.33–1.36), the resonance angles of the reflectance increase gradually. The resonance angle changing with different refractive index of the sensing medium are presented in Figure 4b. In general, the resonance angles of the biosensors increase monotonically with the sensing medium refractive index raise. In contrast, the resonance angle of the biosensor with Au-Si-Au structure is the largest, while that of Al-Si-Al structure is the smallest. The electric field distributions of the biosensor with Au-Si-Au waveguide structure varying with the sensing medium refractive index are shown in Figure 4c. The electric field changes obviously at the PtSe_2_/sensing medium interface, which means that the designed Au-Si-Au based biosensor is sensitive to small changes in the sensing medium refractive index. The tiny changes of sensing medium refractive index near the interface leads to a great alteration of surface wave characteristics, and finally result in the change of electric field. The increase in the sensing medium refractive index leads to the decrease in the biosensor electric field, which also corresponds to the monotonic increase in the reflectance at the resonance angle in Figure 4a. This represents the energy utilization of the biosensor to the incident light is reduced eventually results in the reduction in the internal electric field. Figure 4d shows the sensitivities of the innovative biosensor. Different from the phenomenon that the sensitivity of the biosensors increases monotonically with the raise of the sensing medium refractive index, the sensitivity of the biosensor with Au-Si-Au structure decreases when the sensing medium refractive index is greater than 1.34. This is because the biosensor with Au-Si-Au structure has a very large resonance angle, and the small alteration of the sensing medium refractive index is not enough to make a large offset to the resonance angle. Perhaps compared with other metal waveguides, the biosensor with Au-Si-Au structure has no sensitivity advantage after the refractive index of the sensing medium is greater than 1.34. In general, the proposed structure has excellent detection effect in the range of refractive index of sensing medium from 1.33 to 1.36. FWHM and FOM are two other important parameters of biosensor. In contrast, the biosensor with Au-Si-Au structure has the largest FWHM (Figure 4e), which seriously affected its FOM (Figure 4f) at this refractive indices range.

It is also important to discuss the influence of incident light wavelength on the performance of biosensor. Variations in reflectance in terms of incident angle under incident wavelength from 550 to 750 nm have been shown in Figure 5a. Here, the discussion still uses the BK7-Au-Si-Au-PtSe_2_ (2.0 nm) structure as an example. The reflectance curves become deep and narrow with the red shift of the incident light wavelength, which is contrary to the phenomenon of the increase in the refractive index of the sensing medium. Figure 5b shows the sensitivity curves of the proposed biosensor structure at different wavelengths. Different waveguide metals in biosensors also lead to different response wavelength ranges. The widest response band is the BK7-Al-Si-Al-PtSe_2_ (2.0 nm) structure. In general, the results also show that the proposed biosensor structure can work under different incident light excitation. Figure 5c,d represent FWHM and FOM curves of the proposed biosensor with the change of wavelength, respectively. The FWHM decreases gradually, while the value of FOM increases with the increase in incident light wavelength.

In the following, the thickness of PtSe_2_ layer in the proposed biosensor structure is optimized with the sensing medium refractive index fixed at 1.33. The reflectance curves become broader and shallower rapidly with the increase in PtSe_2_ thickness (Figure 6a). Even the change amplitude is larger than that when the Si layer thickness increases (Figure 4a). This means that the influence of PtSe_2_ thickness on the reflectance of the biosensor is obviously greater than that of Si thickness. However, the resonance angle of the biosensor with Au-Si-Au waveguide structure does not increase monotonically with the PtSe_2_ layer thickness increase from 2 to 7.8 nm, but increases first and then decreases (Figure 6b). This is obviously due to the influence of the larger resonance angle. The electric field distributions of the BK7-Au-Si-Au-PtSe_2_ biosensor with the PtSe_2_ layer thickness increase are shown in Figure 6c. The electric field distributions show an obvious downward trend at the PtSe_2_/sensing medium interface with the increase in PtSe_2_ thickness. Additionally, the value of the electric field at the PtSe_2_/sensing medium interface drops by 96% with the PtSe_2_ layer thickness increasing from 2 to 7.8 nm. This means that the small shift in PtSe_2_ layer thickness has a great impact on the electric field distribution in the biosensor. Figure 6d shows the sensitivity curves of the biosensors with different structures under different thicknesses of PtSe_2_. After comparison, it is found that the BK7-Au-Si-Au-PtSe_2_ (2 nm) biosensor has the highest sensitivity of 193.8°/RIU. In Figure 6, the data only discuss that the thickness of PtSe_2_ increases from 2 to 7.8 nm, because there is only PtSe_2_ dielectric constant in this thickness range in the previous experimental report. If the thickness of PtSe_2_ is reduced to less than 2 nm, the sensitivity of the biosensor structure must change dramatically according to the above discussion. Furthermore, the thickness of PtSe_2_ is different when the biosensors with different structures have the maximum sensitivity. Except for individual points, the overall FWHM of the proposed biosensors shows an upward trend (Figure 6e), while FOM shows a downward trend (Figure 6f) with the increase in the number of PtSe_2_ layers. In general, the FOM of SPR biosensor still has disparity compared with other biosensors [38,39,40]. More parameters are listed in Table 2, which stem from the proposed biosensors. The sensitivity comparison of different PtSe_2_-based biosensors is given in Table 3. After comparison, it is found that the proposed biosensor structure has a significant effect on the enhancement of sensitivity.

## 4. Conclusions

In this work, an innovative biosensor based on metal-Si-metal waveguide structure and PtSe_2_ layer is analyzed and numerically simulated. In the structure, PtSe_2_ is in direct contact with the detection substance as a biomolecular recognition element. At the same time, PtSe_2_ adhered to the metal surface can also protect the metal from oxidation, so as to increase the service life of the biosensor. It is shown that the proposed biosensor structure can detect biomolecules with refractive index of 1.33 to 1.36 and can be driven by different incident light. Through optimization, it is found that the sensitivity of the new biosensor is higher than that of the traditional biosensor. Additionally, it also higher than the traditional biosensor modify with PtSe_2_. Additionally, in the case of BK7-Au-Si-Au-PtSe_2_ (2 nm) biosensor, the maximum is obtained as 193.8°/RIU. This research opens a new gate for the development of biosensors in the future.

## Figures and Tables

**Figure 1 biosensors-12-00027-f001:**
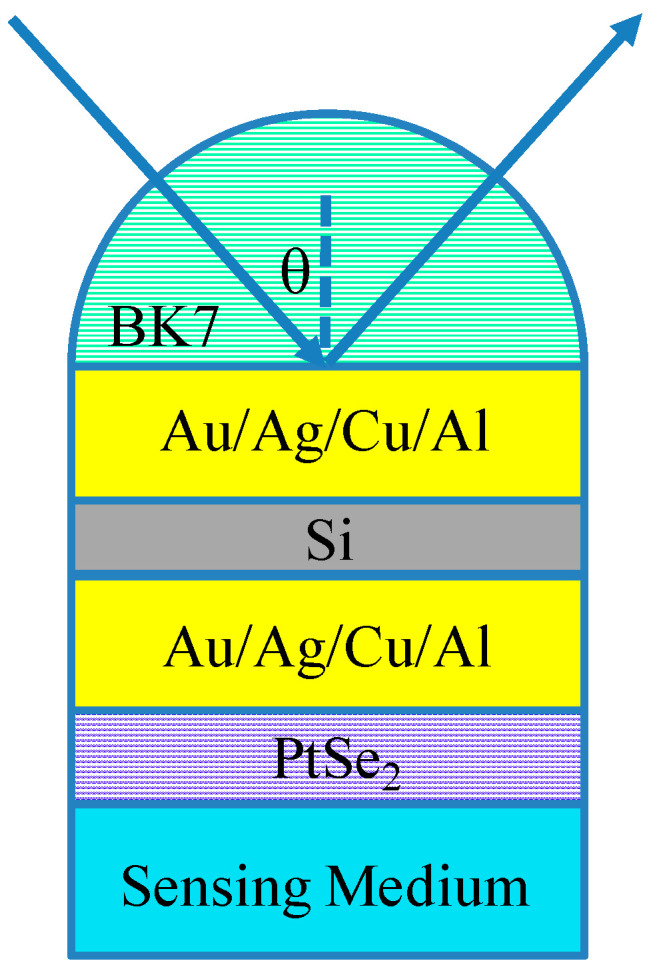
Schematic illustration of the PtSe_2_ SPR biosensor with metal-Si-metal waveguide structure.

**Figure 2 biosensors-12-00027-f002:**
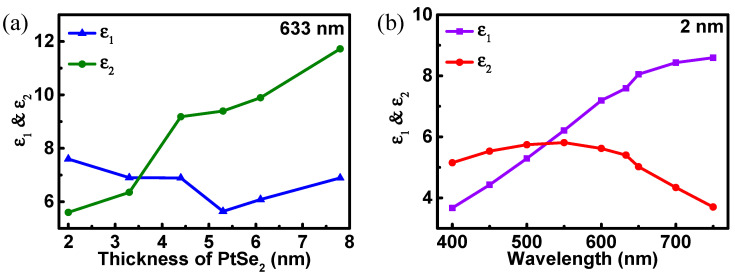
The dielectric constant of PtSe_2_ (**a**) varies over different thickness at wavelength of 633 nm; (**b**) varies over different wavelength at the thickness of 2 nm.

**Figure 3 biosensors-12-00027-f003:**
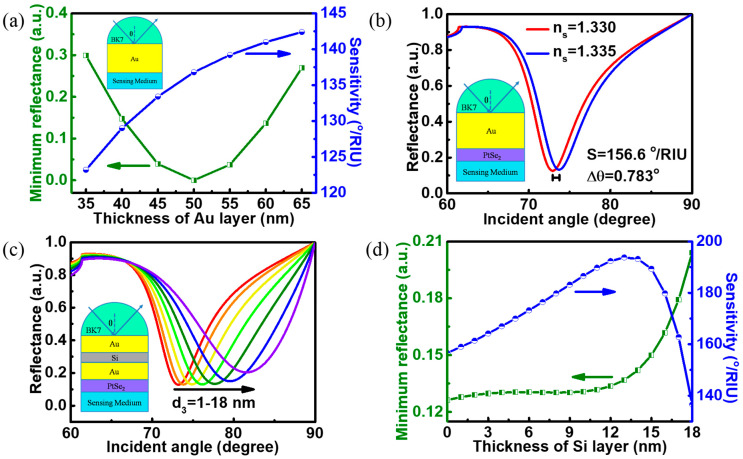
(**a**) Variations in the minimum reflectance and sensitivities of the traditional biosensor with different thicknesses of Au layer; variation in reflectance in terms of incident angle for (**b**) the biosensor with PtSe_2_ on the Au layer; (**c**) the proposed biosensor with PtSe_2_ on the Au-Si-Au waveguide structure (the thicknesses of Si films are 1, 3, 6, 9, 12, 15 and 18 nm); (**d**) variations in the minimum reflectance and sensitivities of the proposed biosensor with different thicknesses of Si layers.

**Figure 4 biosensors-12-00027-f004:**
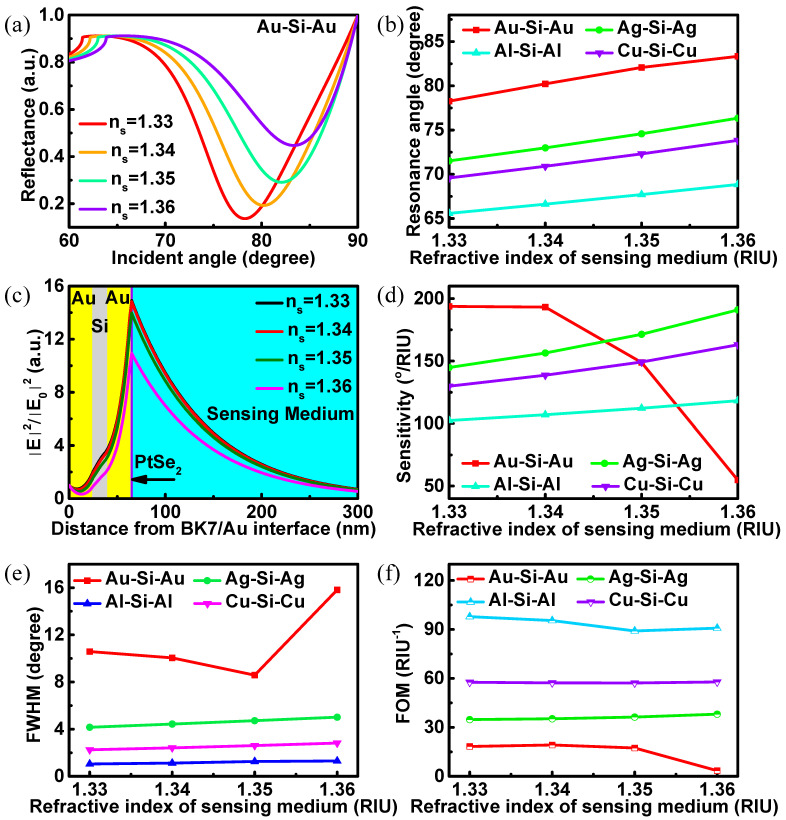
The performance of BK7-metal-Si-metal-PtSe_2_ (2.0 nm) biosensor. (**a**) Variation in reflectance in terms of incident angle (n_s_ = 1.33–1.36); (**b**) variation in the resonance angles of the proposed biosensors with different refractive indices of the sensing medium; (**c**) the electric field distributions of the proposed biosensor (n_s_ = 1.33–1.36); (**d**) the sensitivities; (**e**) the FWHM; (**f**) the FOM as functions of the refractive indices of the sensing medium for the proposed biosensors.

**Figure 5 biosensors-12-00027-f005:**
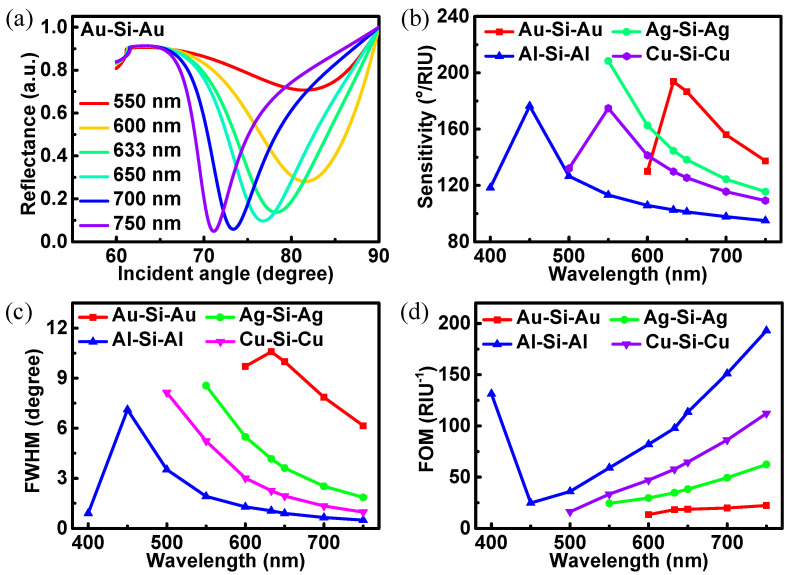
The performance of BK7-metal-Si-metal-PtSe_2_ (2.0 nm) biosensor.(**a**) Variations in reflectance in terms of incident angle (incident wavelength from 550 to 750 nm); (**b**) the sensitivities; (**c**) the FWHM; (**d**) the FOM as functions of the incident wavelength for the proposed biosensors.

**Figure 6 biosensors-12-00027-f006:**
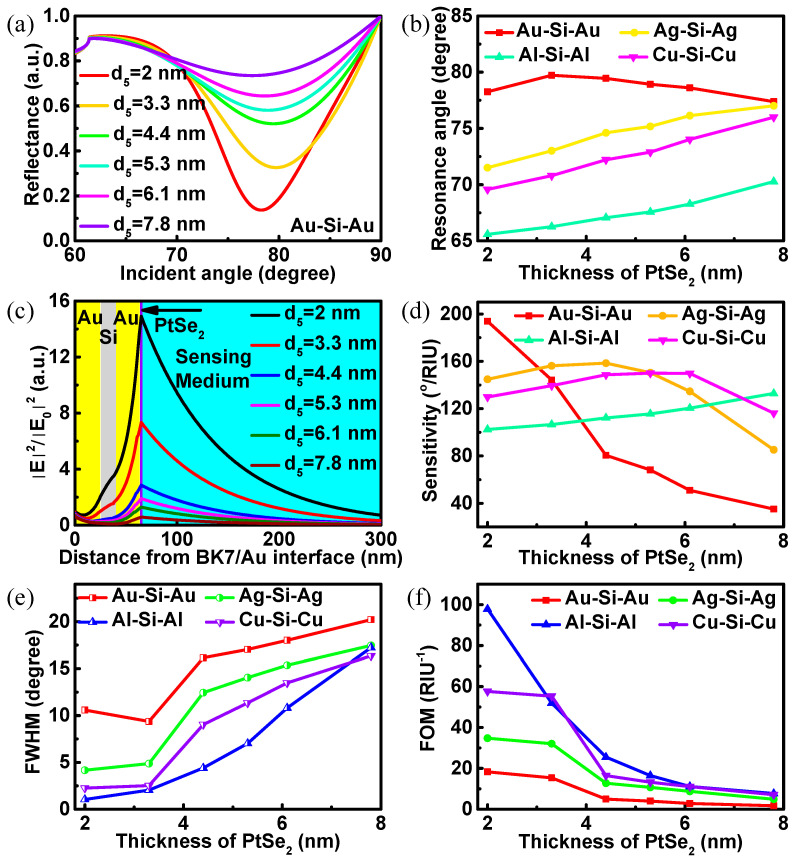
(**a**) The reflectance change in terms of incident angle with different PtSe_2_ thicknesses; (**b**) the resonance angles as functions of PtSe_2_ thicknesses of the proposed biosensors; (**c**) the electric field distributions of the proposed biosensors with the thicknesses of PtSe_2_ range from 2 to 7.8 nm; (**d**) the sensitivities; (**e**) the FWHM; (**f**) the FOM as functions of PtSe_2_ thicknesses of the proposed biosensors.

**Table 1 biosensors-12-00027-t001:** The collision and plasma wavelengths of various metals.

	Au (m)	Ag (m)	Al (m)	Cu (m)
*λ_c_*	8.9342 × 10^−6^	1.7614 × 10^−5^	2.4511 × 10^−5^	4.0852 × 10^−5^
*λ_p_*	1.6826 × 10^−7^	1.4541 × 10^−7^	1.0657 × 10^−7^	1.3617 × 10^−7^

**Table 2 biosensors-12-00027-t002:** Change in different PtSe_2_ thickness, highest sensitivity (S), minimum resonance angle (θ_min_), resonance angle change (∆θ), and the smallest reflectance (R_min_) for the PtSe_2_ SPR biosensors at n_s_ = 1.33.

		PtSe_2_ Thickness (nm)	S (°/RIU)	θ_min_ (degree)	∆θ (degree)	R_min_ (a.u.)	FWHM (degree)	FOM (RIU^−1^)
Au	Without Si and PtSe_2_	0	136.8	70.53	0.684	6.6 × 10^−8^	3.79	36.12
	Without Si and with PtSe_2_	2	156.6	72.92	0.783	0.1263	6.32	24.78
	With Si andPtSe_2_	2	193.8	78.26	0.969	0.1368	10.58	18.31
		3.3	144.2	79.73	0.721	0.3259	9.37	15.39
		4.4	80.6	79.45	0.403	0.5209	16.15	4.99
		5.3	68.2	78.92	0.341	0.5809	17.04	4.00
		6.1	51.0	78.61	0.2550	0.6443	18.01	2.83
		7.8	35.2	77.39	0.176	0.7350	20.21	1.74
Ag	Without Si and PtSe_2_	0	116.0	67.64	0.580	0.0059	1.28	90.63
	Without Si and with PtSe_2_	2	127.4	69.25	0.637	0.1776	2.46	51.84
	With Si andPtSe_2_	2	144.8	71.51	0.724	0.1365	4.16	34.78
		3.3	156.2	73.02	0.781	0.3045	4.87	32.06
		4.4	158.4	74.61	0.792	0.4949	12.44	12.74
		5.3	150.4	75.18	0.752	0.5741	14.05	10.71
		6.1	134.6	76.14	0.673	0.6382	15.36	8.76
		7.8	85.2	77.01	0.426	0.7553	17.46	4.88
Al	Without Si and PtSe_2_	0	95.4	64.37	0.477	0.2542	0.17	570.89
	Without Si and with PtSe_2_	2	100.2	65.21	0.501	0.6658	0.89	112.58
	With Si andPtSe_2_	2	102.4	65.58	0.512	0.6846	1.05	97.80
		3.3	106.6	66.26	0.533	0.7842	2.05	51.80
		4.4	112.2	67.06	0.561	0.8492	4.40	25.49
		5.3	115.6	67.57	0.578	0.8737	7.03	16.45
		6.1	120.4	68.28	0.602	0.8874	10.81	11.14
		7.8	132.8	70.27	0.664	0.9112	17.23	7.71
Cu	Without Si and PtSe_2_	0	109.6	66.69	0.548	0.1143	0.55	200.14
	Without Si and with PtSe_2_	2	119.0	68.06	0.595	0.1953	1.47	80.68
	With Si andPtSe_2_	2	129.8	69.58	0.649	0.1535	2.25	57.65
		3.3	139.4	70.80	0.697	0.3569	2.52	55.33
		4.4	148.6	72.21	0.743	0.5537	9.04	16.45
		5.3	150.0	72.89	0.750	0.6312	11.34	13.22
		6.1	149.8	74.02	0.749	0.6858	13.48	11.12
		7.8	116.2	75.99	0.581	0.7886	16.35	7.11

**Table 3 biosensors-12-00027-t003:** The biosensor structure, incident wavelength (nm), sensitivity (°/RIU) and reference.

Biosensor Structure	Incident Wavelength (nm)	Sensitivity (°/RIU)	Reference
BK7-Au/Ag-PtSe_2_	633	162 (Ag film) 165 (Au film)	[26]
BK7-Au/Ag-PtSe_2_-2D materials	633	194 (Ag/PtSe_2_/WS_2_) 187 (Au/PtSe_2_/WS_2_)	[17]
BAK1-ZnO-silver-PtSe_2_-graphene	633	155.33	[41]
BK7-Au-PtSe_2_-graphene	633	200	[42]
BK7-metal-Si-metal-PtSe_2_	633	193.8	This work

## Data Availability

The data presented in this study are available on request from the corresponding author.

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
