# Peer review of "High-Sensitivity PtSe2 Surface Plasmon Resonance Biosensor Based on Metal-Si-Metal Waveguide Structure"

_biosensors, 2022, doi:10.3390/bios12010027_

Round 1

Reviewer 1 Report

In the manuscript, the authors have proposed an innovative biosensor based on Au-Si-Au waveguide structure and PtSe2 layer. In the structure, PtSe2 is in direct contact with the detection substance as a biomolecular recognition element. The idea of the manuscript seems applicable. There are some issues that need to be addressed: (1)What is the thickness of the Au layer in the design structure? (2)When the refractive index of the material to be measured is greater than 1.34, the sensitivity of the designed structure decreases. Therefore, what are the advantages of the Au-Si-Au waveguide structure compared with other metal waveguide structures? (3)The figure of merit (FOM) value is important to the performance of sensors, and the authors should consider discussing the FOM of the designed structures. (4)The authors should compare the performance with other reported sensors (For examples,Appl. Phys. Express, 2019, 12: 052015, Chin. Phys. B, 2021, 30 (2): 024207,Journal of Lightwave Technology,2021,39(22):7217-7222) to illustrate the advantages of the designed structure.

Reviewer 2 Report

The research proposes a metal-Si-metal waveguide structure to further improve the performance of the biosensor. I would propose some major changes as follows:

  1. There are some problems related to the formatting of the paper. For example, the font size of the paragraph on page 2 (line 49-line 58) is different in comparison with the rest of the text. Moreover, Authors start the new sections with the figures immediately (Section 2; Section 3). I would propose to include the introductory paragraphs before.
  2. Authors should discuss some physical properties of PtSe2.
  3. I think, Authors should stress novelty of their work as a large stream of publications already has been dedicated to the investigations of the waveguides and surface plasmon polaritons. For instance, Authors should cite some recent article in the filed such as
    Eelectrodynamical characteristic particularity of open metamaterial square and circular waveguides, etc.
  4. Authors should include additional plots showing dependencies of the refractive indexes upon wavelength.
  5. I think, it would be desirable to add an additional paragraph to the Introduction describing composition of the rest of the manuscript.

Reviewer 3 Report

Zhitao et al. report on the “High Sensitivity PtSe2 Surface Plasmon Resonance Biosensor Based on Metal-Si-Metal Waveguide Structure.”  The content of the work is interesting, but the manuscript cannot be published in the present form due to the following issues:

  1. The authors have not specified the reason for fixing Au film thickness of 50 nm? What are the consequences of variation of the Au thickness?
  2. In figure 4 (d) the PtSe2 with the thickness 2 nm shows the sensitivity as the highest. An explanation is required. What will happen if the thickness goes lower down than 2 nm.
  3. The author should involve the novelty and the reliability of the system with the other surface plasmon resonance biosensors. This should be added before the conclusion part

Round 2

Reviewer 2 Report

Authors have improved manuscript. I recommend acceptance.